# Short-Term Effects of Cooled Radiofrequency Ablation on Walking Ability in Japanese Patients with Knee Osteoarthritis

**DOI:** 10.3390/jcm13237049

**Published:** 2024-11-22

**Authors:** Kentaro Hiromura, Hironori Kitajima, Chie Hatakenaka, Yoshiaki Shimizu, Terumasa Miyagaki, Masayuki Mori, Kazuhei Nakashima, Atsushi Fuku, Hiroaki Hirata, Yoshiyuki Tachi, Ayumi Kaneuji

**Affiliations:** 1Department of Orthopedic Surgery, Kanazawa Medical University, Kahoku 920-0293, Japan; hiromura@kanazawa-med.ac.jp (K.H.); chie-h@kanazawa-med.ac.jp (C.H.); ysakshmz@kanazawa-med.ac.jp (Y.S.); f-29@kanazawa-med.ac.jp (A.F.); hiro6246@kanazawa-med.ac.jp (H.H.); y-t-s17@kanazawa-med.ac.jp (Y.T.); kaneuji@kanazawa-med.ac.jp (A.K.); 2Department of Orthopedic Surgery, Kanazawa Medical University Himi Municipal Hospital, Himi 935-8531, Japan; 3Department of Rehabilitation, Kanazawa Medical University Himi Municipal Hospital, Himi 935-8531, Japan; miyagaki@kanazawa-med.ac.jp (T.M.); mori@kanazawa-med.ac.jp (M.M.); naka0111@kanazawa-med.ac.jp (K.N.)

**Keywords:** cooled radiofrequency ablation, knee osteoarthritis, walking ability

## Abstract

**Background/Objectives**: Knee osteoarthritis (KOA) is a degenerative joint disease typically managed with conservative treatments, such as anti-inflammatory medications and intra-articular hyaluronic acid injections; however, advanced cases may eventually require surgical intervention. Recently, cooled radiofrequency ablation (CRFA) has emerged as a novel treatment option for alleviating KOA-related pain by temporarily disabling pain-transmitting nerves. This study evaluated the short-term effects of CRFA on pain relief and walking ability in KOA patients, with a specific focus on functional improvements in walking capacity. **Methods**: This study included 58 patients (71 knees) with KOA who underwent CRFA after experiencing inadequate pain control with conservative treatments. The cohort consisted of 28 men and 30 women, with a mean age of 75.2 years (55–90). Under ultrasound guidance, CRFA was performed on the superior lateral geniculate nerve, superior medial geniculate nerve, and inferior medial geniculate nerve, with each targeted nerve ablated. Pre- and post-procedural evaluations (one month after CRFA) included assessments of visual analog scale (VAS) scores for pain at rest and during walking, range of motion (ROM), knee extensor strength, walking speed, and gait stability. **Results**: Significant improvements in the mean VAS (rest/walking) and mean walking speed (comfortable/maximum) were observed following CRFA. However, no significant changes were noted in ROM, knee extensor strength, or walking stability. **Conclusions**: These findings suggest that rehabilitation may be essential to further enhance walking stability. Overall, CRFA appears to be a promising short-term treatment option for reducing VAS pain scores and enhancing walking speed in patients with KOA.

## 1. Introduction

Knee osteoarthritis (KOA) is an irreversible degenerative condition characterized by cartilage degeneration, wear, and deformity or destruction of the bone that results in pain and functional impairment [1,2]. KOA is one of the most common musculoskeletal disorders worldwide and significantly affects the quality of life (QOL), particularly in the elderly population. Due to the aging of the population and rising obesity rates, the prevalence of KOA is increasing, posing a severe public health issue [1,2,3]. The incidence of KOA is influenced by many factors, such as work, sports participation, musculoskeletal injuries, obesity, and gender [4,5]. Furthermore, genetic predisposition has also been identified as a contributing factor, with certain genetic markers associated with a higher likelihood of developing KOA [6]. Biomechanical factors, such as joint misalignment and altered load distribution, can accelerate disease progression by increasing stress on specific areas of the knee joint [7]. In addition, systemic inflammatory conditions, including metabolic syndrome and diabetes, have been linked to an elevated risk of KOA development due to their impact on joint health and cartilage integrity [8]. In cases of mild symptoms and low disease severity, conservative treatments such as oral anti-inflammatory analgesics and intra-articular hyaluronic acid injections are commonly used to manage KOA. However, for patients with more severe symptoms and advanced disease, surgical interventions, such as osteotomy or total knee arthroplasty, are typically employed [1,2,9]. Nevertheless, there is a significant population of patients who are resistant to conservative treatments but for whom surgery is either premature or undesired. This creates a need for treatments that bridge the gap between conservative and surgical options.

In June 2023, cooled radiofrequency ablation (CRFA) using the Coolief^®^ Pain Management System (Avanos Medical, Inc., Georgia, USA) was approved for insurance coverage in Japan as a treatment for KOA patients who experience difficulty in managing pain with conventional conservative therapies. CRFA uses radiofrequency waves to heat nerve tissue, temporarily blocking the function of pain-transmitting nerves around the knee to alleviate discomfort [10,11]. Unlike traditional peripheral nerve radiofrequency ablation (RFA), CRFA features a cooling mechanism that circulates water through the probe tip, preventing tissue charring and allowing for a broader area of nerve ablation [10,11]. This broader ablation zone enables more effective targeting of pain-conducting nerves and may contribute to prolonged pain relief compared with standard RFA techniques. CRFA has been reported to have equivalent safety and greater efficacy compared with intra-articular hyaluronic acid injections and corticosteroid injections [12,13]. Recent studies from various countries have documented significant improvements in pain levels among KOA patients treated with CRFA, underscoring its potential as a reliable alternative for patients who do not achieve sufficient relief with conservative methods [14,15,16].

This study specifically aimed to evaluate the short-term outcomes of CRFA in Japanese patients with KOA, focusing on the degree of pain relief and improvements in knee function and walking ability. No studies have specifically focused on changes in knee function and walking ability following CRFA. By examining knee function, walking ability, and pain levels, this study sought to provide a more comprehensive understanding of how CRFA may benefit KOA patients beyond immediate pain relief.

## 2. Materials and Methods

### 2.1. Participants

The Ethics Committee of Himi City General Hospital, Kanazawa Medical University, approved this retrospective study. Fifty-eight patients (seventy-one knees) with KOA who had experienced inadequate pain control with conventional conservative treatments such as oral anti-inflammatory analgesics and intra-articular hyaluronic acid injections for over six months underwent CRFA and agreed to participate in gait analysis. CRFA was performed by two certified orthopedic surgeons who had completed a training program using the appropriate guidelines set out by the Japanese Society for Joint Diseases. In each case, a test block was performed using 3–5 mL 1% lidocaine injections to the superior lateral geniculate nerve, superior medial geniculate nerve, and inferior medial geniculate nerve under ultrasound guidance, and CRFA was planned for patients who demonstrated an improvement in symptoms by more than 50% (Figure 1). The cohort consisted of 28 men and 30 women, with an average age of 75.2 years (55–90 years) and an average BMI of 25.4 (16.8–39.5). The Kellgren–Lawrence grading (K-L grade) of osteoarthritis was as follows: 16 knees in grade II, 22 in grade III, and 33 in grade IV (Table 1).

The superior lateral genicular nerve, superior medial genicular nerve, and inferior medial genicular nerve were targeted.

### 2.2. CRFA Procedure

All procedures were performed under sterile conditions in the operating room. Patients were supine with a pillow under the knee to maintain slight flexion. During the CRFA, each patient’s heart rate, blood pressure, and oxygen saturation were monitored. The CRFA procedure was conducted using the Coolief^®^ Pain Management Radiofrequency System, which includes a 100 mm, 17-gauge RF introducer with a 4 mm active tip and an 18-gauge cooled RF probe equipped with a saline cooling system. Under ultrasound guidance, ablation sites were identified using the arteries accompanying the superior lateral geniculate nerve, superior medial geniculate nerve, and inferior medial geniculate nerve as anatomical landmarks. Ultrasound-guided nerve identification and probe insertion were performed as described in the existing literature [17]. Approximately 5 mL of a local anesthetic solution (0.75% ropivacaine and 1% lidocaine) was administered at each target site. The cannula was then advanced percutaneously toward the landmarks under ultrasound visualization. Precise nerve localization was achieved through motor stimulation at 1.0 V and 2 Hz to confirm the absence of muscle, followed by sensory stimulation at 50 Hz with a voltage below 0.5 V contractions in the targeted lower limb area. Each targeted nerve was subsequently ablated for 2 min and 30 s at 60 °C using the Coolief^®^ system (Figure 2). To confirm that ablation was performed accurately, the probe was appropriately checked under ultrasound guidance during ablation to ensure it was in the target area. Walking and daily activities were permitted immediately after ablation. We allowed the patients to live without any restrictions on their daily lives and instructed them to return for a follow-up examination in one month.

Under ultrasound guidance, the Coolief^®^ Pain Management Radiofrequency System performed ablation with the superior lateral geniculate nerve, with the CRFA probe located in the introducer needle.

### 2.3. Outcome Measures

The following parameters were assessed and compared at baseline (prior to CRFA) and one-month post-CRFA: visual analog scale (VAS) scores at rest and during ambulation, range of motion (ROM), knee extensor muscle strength, comfortable walking speed, and maximum walking speed. The VAS scores were used to evaluate knee and periarticular pain on a scale from 0 to 10, where 0 indicated no pain, and 10 represented the worst possible pain. ROM was measured in a supine position by a single orthopedic specialist. Knee extensor strength was assessed using a handheld dynamometer (Mobile, Sakai Medical Co., Tokyo, Japan), with three repetitions of maximal isometric contraction and the highest value recorded. Walking speed was determined by timing a 10 m walk with a stopwatch.

Gait analysis was conducted using walkview (HOMER ION Laboratory, Tokyo, Japan), a device equipped with an accelerometer attached to a waist belt, which allows for high-precision gait analysis (Figure 3). The data were collected while subjects walked comfortably and wore waist belts with a walkview device attached. The device was adjusted and used according to the attached instruction manual. Participants were instructed to walk 10 m at a comfortable pace. Measurements were taken in the middle 6 m, with 2 m of space before and after for acceleration and deceleration. This device evaluated the harmonic ratio and gait rhythm. The detailed calculation methods for the harmonic ratio and gait rhythm have been described in the existing literature [18,19,20,21,22]. The same accelerometer was used in the existing research [23]. The harmonic ratio is a composite measure that reflects gait symmetry and smoothness, calculated via frequency analysis of acceleration during walking. It has been reported to correlate with fall risk [20]. Gait rhythm measures the variability in the time taken for each step, which, like the harmonic ratio, has also been associated with fall risk [21].

This device is a gait evaluation device that assesses walking by wearing a belt equipped with sensors around the waist.

### 2.4. Statistical Analysis

The normality of all data was assessed using the Shapiro–Wilk test (*p* > 0.05) [24]. Comparisons of the pre-and post-CRFA data, including VAS scores at rest and during walking, ROM, knee extensor strength, walking speeds, and walkview device parameters, were conducted using either a paired Student’s *t*-test or the Wilcoxon matched-pairs signed-rank test, depending on the data distribution, to evaluate the differences before and one month after the procedure [25]. The walking speed data followed a normal distribution and were analyzed using a paired Student’s *t*-test to compare pre- and post-CRFA measurements. For all other parameters that did not meet the normal distribution, the Wilcoxon matched-pairs signed-rank test was applied to evaluate the differences before and one month after the procedure. All statistical analyses were performed using the GraphPad Prism software (version 9.5.1; GraphPad Software Inc., San Francisco, CA, USA). Statistical significance was accepted at a *p* of <  0.05.

## 3. Results

The mean VAS scores, used to evaluate pain at rest and during walking, demonstrated significant improvements one month after the CRFA procedure. Pre-procedure, the mean VAS score was recorded at 0.96 for resting pain and 6.86 for walking pain, which improved notably to 0.35 at rest and 4.05 during walking at the one-month post-procedure evaluation, indicating a meaningful reduction in pain levels (*p* = 0.011 for resting pain; *p* < 0.001 for walking pain). However, when evaluating the ROM for both extension and flexion of the knee, no statistically significant changes were observed post-procedure, suggesting that CRFA’s impact on joint flexibility may be limited in the short term. Similarly, knee extensor strength showed no significant variation from baseline, indicating that CRFA effectively alleviated pain but did not substantially alter muscle strength within the one-month follow-up period. Meanwhile, the comfortable walking speed saw an increase from 0.91 m per second to 0.99 m per second, and the maximum walking speed improved from 1.30 m per second to 1.34 m per second, with both metrics displaying statistically significant gains (*p* < 0.001 for comfortable speed; *p* = 0.016 for maximum speed). These findings suggest that CRFA has the potential to enhance certain aspects of mobility by increasing walking speed.

Nonetheless, further gait analysis revealed that the harmonic ratio, which assesses gait quality and stability across different directions (anterior–posterior, medial–lateral, and vertical), did not significantly improve. This lack of change extended to other parameters of gait rhythm, indicating that while CRFA may enhance walking speed, it does not necessarily improve gait stability or quality in the short term (Table 2) (Appendix A).

The walking speed data followed a normal distribution and were analyzed using a paired Student’s *t*-test to compare pre- and post-CRFA measurements. For all other parameters that did not meet the normal distribution, the Wilcoxon matched-pairs signed-rank test was applied to evaluate the differences before and one month after the procedure.

Additionally, no significant adverse events were noted in this patient group; notably, there were no reported cases of complications, such as burns, infections, nerve paralysis, or accelerated progression of osteoarthritis, suggesting that CRFA appears to be a relatively safe intervention within this observational period.

## 4. Discussion

Numerous studies have highlighted the effectiveness of CRFA in relieving pain associated with KOA, supporting its growing role as an alternative treatment for this condition [14,15,16]. In our study, which focused on Japanese patients with KOA, we similarly observed significant pain reduction after CRFA, which aligns with previous research on this treatment approach. Notably, our patient cohort had a relatively high mean age of 75.2 years, which emphasizes the applicability of CRFA to older individuals who are often at higher risk of complications from surgical interventions like total knee arthroplasty (TKA). For many elderly patients, particularly those with complex medical conditions or comorbidities, TKA may present significant surgical risks, making CRFA a potentially safer alternative for pain management. Additionally, a substantial number of these older patients prefer to avoid TKA, either due to personal preferences or concerns about recovery and potential complications. For these individuals, CRFA provides an effective pain management option that does not carry the same risks or invasiveness as surgery, making it a viable and attractive choice within this population. Thus, our findings support CRFA as a valuable treatment alternative that can deliver pain relief comparable to that reported in younger cohorts.

At the one-month postoperative follow-up, ROM in both extension and flexion remained limited, indicating no substantial improvement. This outcome may be explained by the fact that CRFA targets pain relief through nerve ablation but does not directly address the underlying anatomical abnormalities or structural deformities of the knee joint, such as cartilage degradation, bone changes, or ligamentous instability. These anatomical issues are vital contributors to limited ROM in KOA patients, and without interventions aimed at modifying joint structure, substantial improvements in ROM may not be achievable. Additionally, the observed lack of progress in knee extensor strength could be due to the relatively brief one-month follow-up period, which might not be sufficient to allow for observable gains in muscle strength. Furthermore, restricted ROM could hinder patients’ ability to effectively strengthen their knee extensor muscles, as limited movement can reduce the range in which muscles can work optimally. Therefore, CRFA effectively addresses pain but does not resolve joint deformities or muscular limitations. It suggests that a more extended follow-up period and potential additional interventions may be necessary to observe any meaningful gains in ROM and knee extensor strength.

When examining walking ability, both comfortable and maximum walking speeds showed significant improvements following the reduction in pain. It is known that KOA patients often develop compensatory movements to avoid pain during walking, leading to reduced walking speed [26]. In this study, CRFA alleviated pain, eliminating the need for compensatory actions, which likely contributed to increased walking speed. Improved walking speed has been associated with enhanced QOL and functional capacity [27]. Moreover, it has been reported that KOA patients with decreased walking speed face an increased mortality risk [28]. Conversely, improvements in walking speed may reduce the risk of chronic conditions such as cerebrovascular disease and dementia [29]. Therefore, enhanced walking speed is essential for long-term health maintenance in KOA patients.

Meanwhile, improvements in walking stability, as assessed by the harmonic ratio and gait rhythm, were not observed one month after CRFA. Despite the improvements in pain and walking speed, several factors may have contributed to the lack of improvement in walking stability. First, KOA patients often experience reduced lower limb strength due to decreased physical activity caused by pain. In particular, the weakening of the quadriceps and hamstrings has been reported to contribute to decreased walking stability [30]. Even if CRFA alleviates pain, walking stability may not improve without concurrent recovery of muscle strength. Second, KOA patients are reported to have reduced proprioception due to receptor abnormalities caused by damage to intra-articular structures such as cartilage and ligaments [31]. When proprioception is diminished, the ability to accurately sense joint movement is impaired, which may lead to decreased walking stability [32]. Since CRFA does not directly address receptor function or proprioception, this could explain the lack of improvement in walking stability. Moreover, while there is a theoretical possibility that CRFA could increase walking instability due to nerve damage caused by ablation, no increase in walking instability was observed in this study. However, it cannot be denied that increasing gait speed without improving gait stability may increase the risk of falls.

Therefore, to improve walking stability, it is advisable to engage in active rehabilitation while pain is alleviated by CRFA, aiming to increase muscle strength and stabilize gait. Looking ahead, further studies should explore the integration of CRFA with comprehensive rehabilitation programs, combining pain relief with interventions targeting muscle strengthening and proprioception enhancement. This multimodal approach could amplify the benefits of CRFA, addressing pain and functional impairments such as limited ROM and walking instability. Improving functional impairment contributes to enhanced QOL. Improving walking stability could lead to fall prevention, extended healthy life expectancy, and reduced healthcare costs [33,34]. While CRFA is an effective treatment for alleviating pain in KOA, further investigation is needed regarding its impact on functional impairments such as limited ROM and walking instability.

This study has several limitations. As a retrospective study with a relatively small sample size (71 knees), the ability to establish causality and the statistical power of the findings are limited. Additionally, this study has limitations that may affect the generalizability of its findings, including the lack of a control group and the focus on a Japanese cohort. We aimed to evaluate immediate functional changes, and based on this, we set 1 month as the initial evaluation time point. However, because it was difficult to adequately observe changes in the ROM and muscle strength during this period, we emphasize that future studies should include a more extended follow-up period. However, this study provides important preliminary data on the short-term effects of CRFA, mainly since the treatment was only recently approved for insurance coverage in Japan. Future studies with larger sample sizes, extended follow-up periods, and randomized controlled trials involving more extensive and diverse populations are essential to confirm these findings, evaluate the durability of CRFA’s benefits, and enhance its functional outcomes. It is also necessary to demonstrate the possibility that a treatment strategy that combines CRFA and rehabilitation may increase therapeutic outcomes.

## 5. Conclusions

This study demonstrated that CRFA effectively provided pain relief, consistent with previous reports, even in a relatively older cohort of KOA patients. While pain relief and increased walking speed were achieved, improvements in walking stability, ROM, and knee extensor strength were not observed at the one-month follow-up. Rehabilitation may be essential for improving functional impairments such as limited ROM, walking instability, and weakness in knee extensor strength. CRFA appears to be a useful short-term treatment option for improving VAS scores and walking speed in KOA patients. These findings highlight the need for further investigation into the long-term effects of CRFA and its potential role in comprehensive treatment strategies for KOA patients.

## Figures and Tables

**Figure 1 jcm-13-07049-f001:**
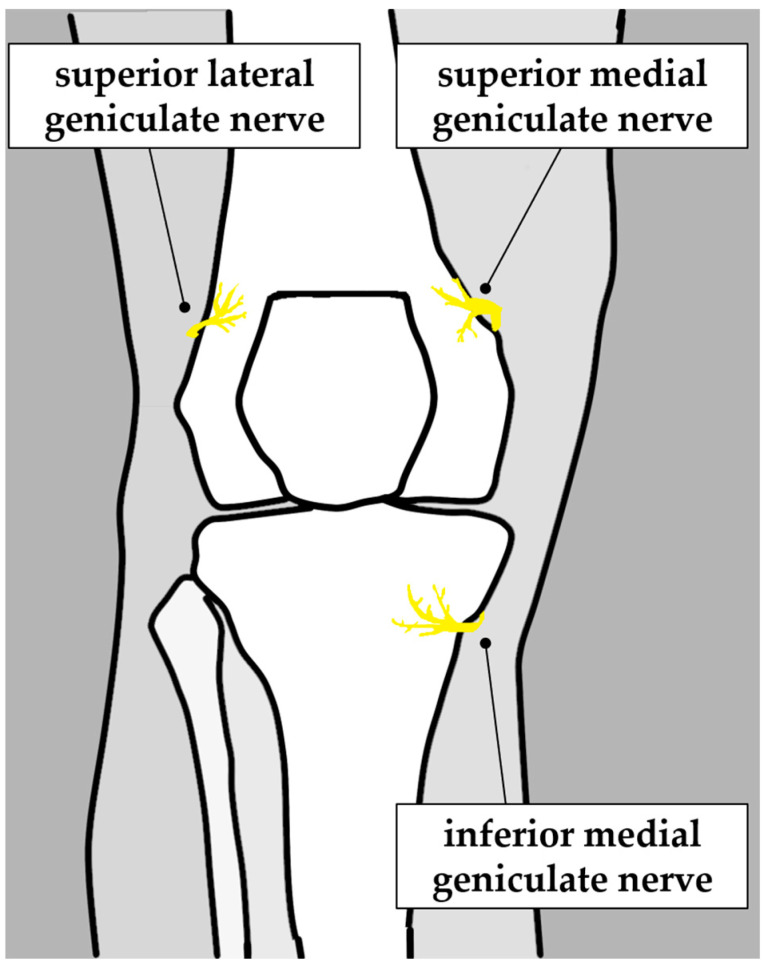
CRFA target nerves.

**Figure 2 jcm-13-07049-f002:**
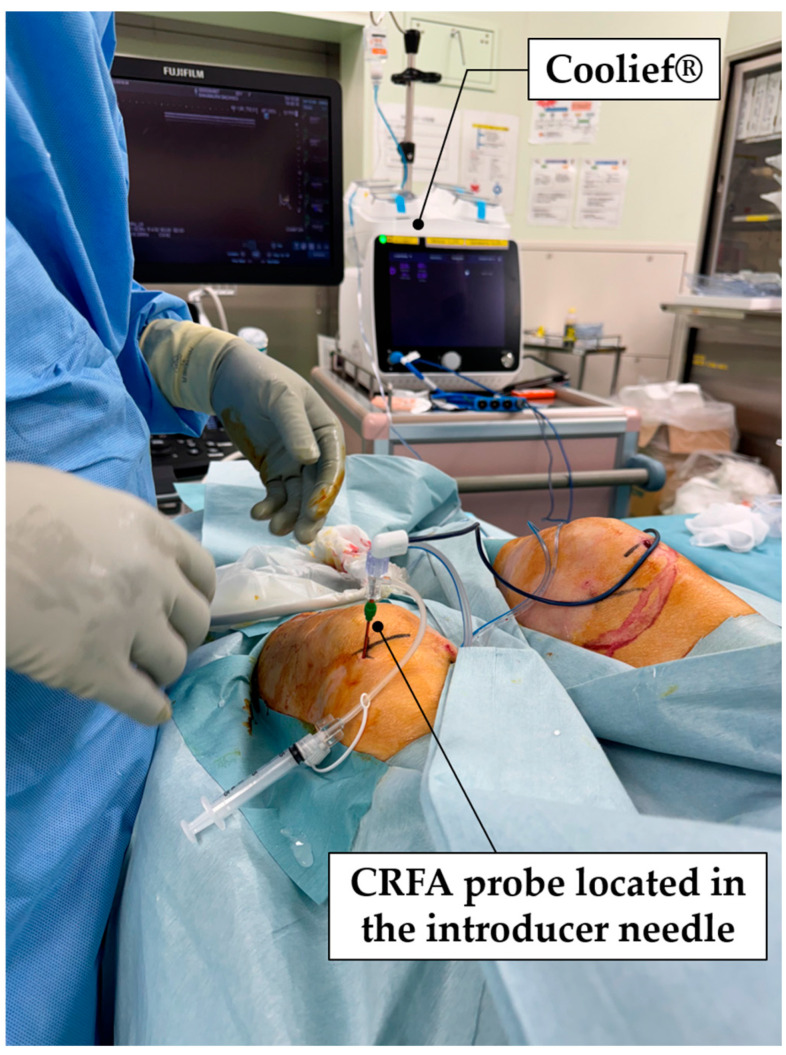
Treatment of CRFA with Coolief^®^ Pain Management Radiofrequency System (Avanos Medical, Inc., Alpharetta, GA, USA).

**Figure 3 jcm-13-07049-f003:**
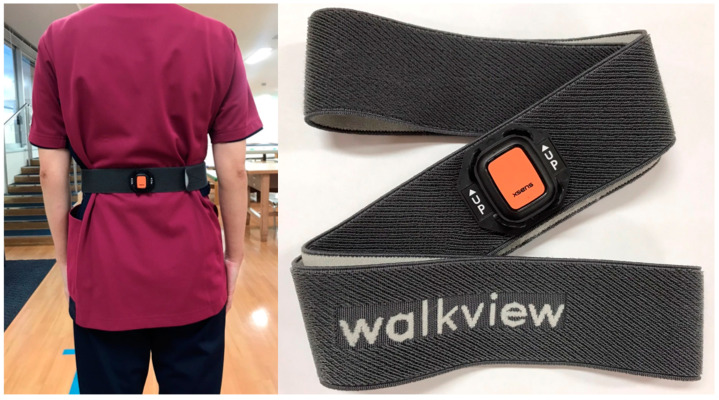
walkview device.

**Table 1 jcm-13-07049-t001:** Patient demographics.

Age at CRFA (y, range)	75.2 (55–90)
Gender (n)	Male: 28; female: 30
BMI (kg/m^2^, range)	25.4 (16.8–39.5)
K-L grade (1, 2, 3, 4)	(0, 16, 22, 33)

**Table 2 jcm-13-07049-t002:** The changes in scores before and after CRFA.

	Pre-CRFA	One Month After CRFA	*p*-Value
Visual analog scale (VAS)			
Resting VAS (range)	0.96 (0–7.4)	0.35 (0–6)	***p* = 0.011**
Walking VAS (range)	6.86 (2.2–10)	4.05 (0–9.3)	***p* < 0.001**
Range of motion (ROM)			
ROM extension (°, range)	−1.62 (−10–0)	−1.62 (−10–0)	*p* = 0.929
ROM flexion (°, range)	124.6 (90–145)	126.2 (90–145)	*p* = 0.080
Knee extension strength (kgf, range)	23.0 (1.9–60.8)	23.1 (4.7–70.3)	*p* = 0.784
Walking speed (WS)			
Comfortable WS (m/s, range)	0.91 (0.32–1.46)	0.99 (0.39–1.53)	***p* < 0.001**
Maximum WS (m/s, range)	1.30 (0.43–2.23)	1.34 (0.54–2.06)	***p* = 0.016**
Harmonic ratio (HR)			
Total HR (range)	85.2 (69–94)	84.1 (54–95)	*p* = 0.640
Anterior and posterior HR (range)	82.8 (55–96)	82.5 (39–97)	*p* = 0.880
Medial–lateral HR (range)	77.1 (57–91)	76.6 (48–91)	*p* = 0.979
Up and down HR (range)	83.9 (53–94)	82.8 (49–95)	*p* = 0.583
Gait rhythm (range)	86.2 (10–97)	84.6 (10–97)	*p* = 0.262

## Data Availability

The authors declare that the data supporting the findings of this study are available upon request.

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
