# Peer review of "Short-Term Effects of Cooled Radiofrequency Ablation on Walking Ability in Japanese Patients with Knee Osteoarthritis"

_jcm, 2024, doi:10.3390/jcm13237049_

Round 1
Reviewer 1 Report
Comments and Suggestions for Authors
General characteristics and evaluation of the reviewed article Short-Term Effects of Cooled Radiofrequency Ablation on Walking Ability in Japanese Patients with Knee Osteoarthritis:
The manuscript addresses a significant clinical issue by evaluating the short-term effects of cooled radiofrequency ablation (CRFA) in patients with knee osteoarthritis (KOA) who did not respond to conservative treatments. The study demonstrates that CRFA significantly reduces pain (as measured by VAS) and improves walking speed one month post-procedure, highlighting its potential as a short-term intervention for pain relief and mobility enhancement.
Strengths of the study include its focus on a novel treatment approach, a clearly defined cohort, and the use of ultrasound-guided nerve ablation for procedural precision. The evaluation of clinically relevant outcomes, such as pain, walking speed, range of motion (ROM), knee extensor strength, and walking stability, provides a comprehensive analysis of CRFA’s effects.
However, the study is limited by the absence of a control group, which weakens causal inferences, and a short follow-up period, restricting insights into the durability of CRFA’s benefits. Additionally, while pain relief and walking speed improved, there were no significant changes in ROM, knee extensor strength, or walking stability, emphasizing the need for adjunct rehabilitation. The study's focus on a Japanese cohort may also limit generalizability to other populations.
In conclusion, this well-executed study demonstrates CRFA’s promise as a short-term treatment for KOA-related pain and mobility issues. Future research with longer follow-up periods, control groups, and diverse patient populations is recommended to strengthen the evidence and explore multimodal treatment strategies.
The article is interesting, addresses a timely and important topic and definitely fits the scope of the journal. It is written generally correctly and requires only minor corrections and additions. Below are my points and detailed comments.
Minor comments:
The abstract is too long and does not meet the editorial requirements of the journal. Please adjust the abstract to the requirements of the journal AND present only the most important information about the article in it.
The introduction is far too short and does not present the problem in sufficient detail. Furthermore, the authors cite only 8 papers in this section, which is significantly too few for a scientific article. Please expand the introduction with more detailed information on osteoarthritis and add the latest literature.
Please expand on the osteoarthritis information in the introduction. This will allow for a better introduction to the topic and highlight the importance of the problem. The incidence of osteoarthritis is influenced by many factors, such as work, sports participation, musculoskeletal injuries, obesity, and gender. Information about these factors, along with relevant literature, should be added to the first paragraph of the introduction. I suggest adding the following references to this paragraph:
https://doi.org/10.3390/healthcare12161648
DOI: 10.1056/NEJMcp1903768
Please mark in bold the results showing statistically significant differences at the assumed level of significance in the tables in Chapter 3. This will facilitate the interpretation of the results.
The statistical analysis section requires expansion and the addition of relevant literature concerning the statistical tests used in the analyses.
In the final part of the discussion, please describe in more detail the limitations of the proposed method, the simplifications used, and a proposal for solving them in the authors' further planned future research.
Once the appropriate corrections and additions have been made, the paper can be further processed. I congratulate the authors on the interesting paper and wish them further success.
Author Response
The manuscript addresses a significant clinical issue by evaluating the short-term effects of cooled radiofrequency ablation (CRFA) in patients with knee osteoarthritis (KOA) who did not respond to conservative treatments. The study demonstrates that CRFA significantly reduces pain (as measured by VAS) and improves walking speed one month post-procedure, highlighting its potential as a short-term intervention for pain relief and mobility enhancement.
Strengths of the study include its focus on a novel treatment approach, a clearly defined cohort, and the use of ultrasound-guided nerve ablation for procedural precision. The evaluation of clinically relevant outcomes, such as pain, walking speed, range of motion (ROM), knee extensor strength, and walking stability, provides a comprehensive analysis of CRFA’s effects.
However, the study is limited by the absence of a control group, which weakens causal inferences, and a short follow-up period, restricting insights into the durability of CRFA’s benefits. Additionally, while pain relief and walking speed improved, there were no significant changes in ROM, knee extensor strength, or walking stability, emphasizing the need for adjunct rehabilitation. The study's focus on a Japanese cohort may also limit generalizability to other populations.
In conclusion, this well-executed study demonstrates CRFA’s promise as a short-term treatment for KOA-related pain and mobility issues. Future research with longer follow-up periods, control groups, and diverse patient populations is recommended to strengthen the evidence and explore multimodal treatment strategies.
The article is interesting, addresses a timely and important topic and definitely fits the scope of the journal. It is written generally correctly and requires only minor corrections and additions. Below are my points and detailed comments.
We thank the reviewer for their thorough assessment and constructive feedback on our manuscript. Below, we address each of the comments raised and provide details on the revisions made to the manuscript.
Minor comments:
The abstract is too long and does not meet the editorial requirements of the journal. Please adjust the abstract to the requirements of the journal AND present only the most important information about the article in it.
Response: Thank you for providing your kind comments. The abstract has been shortened to 250 words or less to comply with journal requirements.
The introduction is far too short and does not present the problem in sufficient detail. Furthermore, the authors cite only 8 papers in this section, which is significantly too few for a scientific article. Please expand the introduction with more detailed information on osteoarthritis and add the latest literature.
Please expand on the osteoarthritis information in the introduction. This will allow for a better introduction to the topic and highlight the importance of the problem. The incidence of osteoarthritis is influenced by many factors, such as work, sports participation, musculoskeletal injuries, obesity, and gender. Information about these factors, along with relevant literature, should be added to the first paragraph of the introduction. I suggest adding the following references to this paragraph:
https://doi.org/10.3390/healthcare12161648
DOI: 10.1056/NEJMcp1903768
Response: Thank you for your useful suggestions. We acknowledge the brevity of the original introduction and have expanded it to include more detailed information on osteoarthritis (OA). The latest literature, including the suggested references (DOI: 10.3390/healthcare12161648 and DOI: 10.1056/NEJMcp1903768), has been incorporated to provide a comprehensive background and emphasize the significance of the problem.
Line47-55
“The incidence of KOA is influenced by many factors, such as work, sports participation, musculoskeletal injuries, obesity, and gender [4.5]. Furthermore, genetic predisposition has also been identified as a contributing factor, with certain genetic markers associated with a higher likelihood of developing KOA [6]. Biomechanical factors, such as joint misalignment and altered load distribution, can accelerate disease progression by increasing stress on specific areas of the knee joint [7]. In addition, systemic inflammatory conditions, including metabolic syndrome and diabetes, have been linked to an elevated risk of KOA development due to their impact on joint health and cartilage integrity [8].”
Please mark in bold the results showing statistically significant differences at the assumed level of significance in the tables in Chapter 3. This will facilitate the interpretation of the results.
Response: Thank you for providing your kind comments. Following your suggestion, we have marked statistically significant results in bold in all relevant tables in Chapter 3 to enhance clarity and facilitate interpretation.
The statistical analysis section requires expansion and the addition of relevant literature concerning the statistical tests used in the analyses.
Response: Thank you for providing your kind comments. We have expanded the statistical analysis section to include a more detailed explanation of the tests used and their appropriateness for the data distribution. Additionally, we have included references to relevant literature to substantiate our methodology.
Line168-176
“The normality of all data was assessed using the Shapiro-Wilk test (p > 0.05) [24]. Comparisons of pre-and post-CRFA data, including VAS scores at rest and during walking, ROM, knee extensor strength, walking speeds, and walkview device parameters, were conducted using either a paired Student’s t-test or the Wilcoxon matched-pairs signed-rank test, depending on the data distribution, to evaluate the differences before and one month after the procedure [25]. Walking speed data followed a normal distribution and were analyzed using a paired Student’s t-test to compare pre-and post-CRFA measurements. For all other parameters that did not meet the normal distribution, the Wilcoxon matched-pairs signed-rank test was applied to evaluate the differences before and one month after the procedure.”
In the final part of the discussion, please describe in more detail the limitations of the proposed method, the simplifications used, and a proposal for solving them in the authors' further planned future research.
Response: Thank you for your useful suggestions. We have elaborated on the limitations of our study, including the absence of a control group, the short follow-up period, and the focus on a Japanese cohort, which may affect the generalizability of our findings. We have also outlined potential solutions and future research directions, such as conducting randomized controlled trials with more extensive and diverse populations and integrating rehabilitation programs to improve functional outcomes.
Line285-297
“Additionally, this study has limitations that may affect the generalizability of its findings, including the lack of a control group and the focus on a Japanese cohort. We aimed to evaluate immediate functional changes, and based on this, we set 1 month as the initial evaluation time point. However, because it is difficult to adequately observe changes in the ROM and muscle strength during this period, we emphasize that future studies should include a more extended follow-up period. However, this study provides important preliminary data on the short-term effects of CRFA, mainly since the treatment was only recently approved for insurance coverage in Japan. Future studies with larger sample sizes, extended follow-up periods, and randomized controlled trials in-volving more extensive and diverse populations are essential to confirm these findings, evaluate the durability of CRFA's benefits, and enhance functional outcomes. It is also necessary to demonstrate the possibility that a treatment strategy that combines CRFA and rehabilitation may increase therapeutic outcomes.”
Once the appropriate corrections and additions have been made, the paper can be further processed. I congratulate the authors on the interesting paper and wish them further success.
Reviewer 2 Report
Comments and Suggestions for Authors
I appreciated the opportunity to review this paper that evaluates the short-term effects of Cooled Radiofrequency Ablation on pain relief and walking ability in Japanese patients with Knee Osteoarthritis who were unresponsive to conservative treatments by analyzing 58 patients for a total of 71 knees.
I just have few comments that I think could improve the manuscript:
- There is the need to better explain the consistency of gait analysis and the calibration of equipment used for measuring walking parameters
- the introduction could include a brief explanation of how the technique object of the manuscript compares to other nonsurgical interventions (for example intra-articular injections) in terms of efficacy and side effects.
- why were a one-month follow-up chosen? Is this timeframe sufficient to observe changes in ROM and muscle strength?
- the authors should consider adding bar charts to illustrate changes in pain scores, walking speed, and other parameters
- The authors should expand on the implications of increased walking speed, for example higher risk of falls, without any improvements in stability
Author Response
I appreciated the opportunity to review this paper that evaluates the short-term effects of Cooled Radiofrequency Ablation on pain relief and walking ability in Japanese patients with Knee Osteoarthritis who were unresponsive to conservative treatments by analyzing 58 patients for a total of 71 knees.
We thank the reviewer for their thorough assessment and constructive feedback on our manuscript. Below, we address each of the comments raised and provide details on the revisions made to the manuscript.
I just have few comments that I think could improve the manuscript:
- There is the need to better explain the consistency of gait analysis and the calibration of equipment used for measuring walking parameters
Response: Thank you for your useful suggestions. We appreciate your bringing this important point to our attention. We have expanded the Methods section of the article to include how we calibrated the Walk View system and the steps we took to ensure consistency in our gait analysis.
Line151-158
“The data was collected while subjects walked comfortably and wore waist belts with a walkview device attached. The device was adjusted and used according to the attached instruction manual. Participants were instructed to walk 10 meters at a comfortable pace. Measurements were taken in the middle 6 meters, with 2 meters of space before and after for acceleration and deceleration. This device evaluated the Harmonic Ratio and gait rhythm. Detailed calculation methods for Harmonic Ratio and gait rhythm are described in existing literature [18-22]. The same accelerometer is used in existing research [23].”
- the introduction could include a brief explanation of how the technique object of the manuscript compares to other nonsurgical interventions (for example intra-articular injections) in terms of efficacy and side effects.
Response: Thank you for your useful suggestions. In the introduction section, we compared CRFA with intra-articular hyaluronic acid injections and Corticosteroid Injections, a traditional conservative treatment, and briefly described its effectiveness and side effects.
Line74-75
“CRFA has been reported to have equivalent safety and greater efficacy compared to intra-articular hyaluronic acid injections and Corticosteroid Injections [12,13].”
- why were a one-month follow-up chosen? Is this timeframe sufficient to observe changes in ROM and muscle strength?
Response: Thank you for your useful suggestions. We aimed to evaluate immediate functional changes, and based on this, we set 1 month as the initial evaluation time point. However, because it is difficult to adequately observe changes in the ROM and muscle strength during this period, we emphasize this as a limitation and suggest that future studies should include a longer follow-up period.
Line285-297
“Additionally, this study has limitations that may affect the generalizability of its findings, including the lack of a control group and the focus on a Japanese cohort. We aimed to evaluate immediate functional changes, and based on this, we set 1 month as the initial evaluation time point. However, because it is difficult to adequately observe changes in the ROM and muscle strength during this period, we emphasize that future studies should include a more extended follow-up period. However, this study provides important preliminary data on the short-term effects of CRFA, mainly since the treatment was only recently approved for insurance coverage in Japan. Future studies with larger sample sizes, extended follow-up periods, and randomized controlled trials in-volving more extensive and diverse populations are essential to confirm these findings, evaluate the durability of CRFA's benefits, and enhance functional outcomes. It is also necessary to demonstrate the possibility that a treatment strategy that combines CRFA and rehabilitation may increase therapeutic outcomes.”
- the authors should consider adding bar charts to illustrate changes in pain scores, walking speed, and other parameters
Response: Thank you for providing your kind comments. Following your suggestions for visual aids to more clearly present the results, we have added bar graphs to show the change in critical results. These figures have been incorporated into the Supplementary Materials section and are referenced appropriately in the text.
- The authors should expand on the implications of increased walking speed, for example higher risk of falls, without any improvements in stability
Response: Thank you for your useful suggestions. We have expanded the "Discussion" section in response to your comment about the impact of increasing walking speed without improving gait stability. In particular, we pointed out that increasing walking speed without improving gait stability may increase the risk of falls. We proposed that a treatment strategy combining CRFA and rehabilitation is necessary to mitigate this risk.
Line270-274
“However, it cannot be denied that increasing gait speed without improving gait stability may increase the risk of falls. Therefore, to improve walking stability, it is advisable to engage in active rehabilitation while pain is alleviated by CRFA, aiming to increase muscle strength and stabilize gait.”